# The Road More Travelled: The Differential Effects of Spatial Experience in Young and Elderly Participants

**DOI:** 10.3390/ijerph18020709

**Published:** 2021-01-15

**Authors:** Antonella Lopez, Alessandro Germani, Luigi Tinella, Alessandro Oronzo Caffò, Albert Postma, Andrea Bosco

**Affiliations:** 1Department of Educational Sciences, Psychology, Communication, University of Bari, 70121 Bari, Italy; luigi.tinella@uniba.it (L.T.); alessandro.caffo@uniba.it (A.O.C.); andrea.bosco@uniba.it (A.B.); 2Department of Philosophy, Social Sciences and Education, University of Perugia, 06123 Perugia, Italy; alessandro.germani@unipg.it; 3Helmholtz Institute, Experimental Psychology, Utrecht University, 3584 CS Utrecht, The Netherlands; A.Postma@uu.nl

**Keywords:** categorical and coordinate spatial relations, sketch map, familiar environments, cognitive map, navigation, spatial experience, environmental affordance, age

## Abstract

Our spatial mental representations allow us to give refined descriptions of the environment in terms of the relative locations and distances between objects and landmarks. In this study, we investigated the effects of familiarity with the everyday environment, in terms of frequency of exploration and mode of transportation, on categorical and coordinate spatial relations, on young and elderly participants, controlling for socio-demographic factors. Participants were tested with a general anamnesis, a neuropsychological assessment, measures of explorations and the Landmark Positioning on a Map task. The results showed: (a) a modest difference in performance with categorical spatial relations; (b) a larger difference in coordinate spatial relations; (c) a significant moderating effect of age on the relationship between familiarity and spatial relations, with a stronger relation among the elderly than the young. Ceteris paribus, the role of direct experience with exploring their hometown on spatial mental representations appeared to be more important in the elderly than in the young. This advantage appears to make the elderly wiser and likely protects them from the detrimental effects of aging on spatial mental representations.

## 1. Introduction

The study of human spatial cognition is concerned with how people acquire and use knowledge about their environment to determine where they are, where to obtain goods and services, and how to find their way home [1]. According to Montello and Raubal [2] (p. 250), the study of spatial cognition can also be defined as *the area of research that studies activities centrally involving explicit mental representation of space,* such as everyday tasks involving navigation and wayfinding, as well as producing and interpreting symbolic depictions and spatial language. The ability to build spatial mental representations allows humans to describe their environments in detail, both in terms of their spatial attributes and, more specifically, in terms of the relative locations and distances between objects and landmarks (e.g., [3,4,5]). Spatial mental representations are internal representations of environmental features and their spatial relations (e.g., [6]). These internal images of the surrounding environment gradually consolidate through experience [7]. In order to maintain accurate spatial mental representations, humans may rely on topographical knowledge, that is, their knowledge of landmarks and the relations between them.

In the present study, we address three main aspects of spatial cognition. First, we discuss two types of relations that play a crucial role in spatial mental representations: categorical and coordinate spatial relations. Categorical relations specify the relative positions of objects (above or below, on the left or on the right) and inter-object distances (near to or far from). Then, we examine the role of familiarity with a given environment. Familiarity is thought to reflect both the amount of time spent in a place and how well people, subjectively, think they know that place. In the present study, the concept of familiarity is integrated with the concept of affordance, in terms of possibilities for action. Finally, we examine several intervening factors that have an impact on the relationship between familiarity with an environment and spatial mental representations, with a special focus on aging. The core issue addressed by this study is how much familiarity with the hometown environment affects spatial mental representations of that environment. More importantly, we consider if familiarity is moderated by the age of the participants, even holding other potentially interfering variables—residential zones, socio-economic status, gender, education, and cognitive efficiency/intelligence—constant. In line with the general purposes of the present study, we operationalize familiarity as the frequency of exploration opportunities and the degree of activity/autonomy enjoyed in this exploration. Since cognitive decline may take different trajectories in relation to the two types of spatial mental representations or information, that is, categorical positions and coordinate distances, we compute two distinct statistical models. Based on our evidence and analyses, we suggest that the quality and quantity of experience with an environment have differential effects and outcomes in relation to age.

### 1.1. The Building Blocks of Spatial Mental Representations

People most commonly learn topographical information through direct exploration of the environment. This enables them to process spatial relations in conjunction with other relevant features. Topographical knowledge is not stored as a unitary representation [8]. For instance, throughout their life, people use *landmark knowledge*, information about the location of salient and prominent topographical elements present in the environment, *route/procedural knowledge*, information gleaned from exploration of the environment, and *survey/configurational knowledge*, more general knowledge that reflects the absolute positions of landmarks [9,10]. Landmarks are encoded in two different spatial frames of reference: an egocentric frame encodes spatial information about the location of the agent in the environment, and an allocentric frame encodes spatial information about the position of objects relative to each other, independent of individual body space (for a review, see [11]). The acquisition of spatial knowledge about the environment was originally thought to ascend a hierarchy of levels from landmark knowledge (initial level) to route/procedural knowledge (intermediate level), and finally to survey/configurational knowledge (advanced level) [12]. Later studies, however, suggested that survey knowledge is not acquired after, but rather simultaneously with route knowledge [13,14]. Whatever the level of representation, two types of spatial relations play a crucial role: *categorical and coordinate spatial relations*, which, respectively, specify relative inter-object positions (above or below, on the left or on the right) and inter-object distances (near to or far from). According to Kosslyn [15], an accurate encoding of categorical and coordinate spatial relations is an essential prerequisite for navigation and the construction of spatial mental representations. Hence, in addition to frames of reference (i.e., egocentric, allocentric) and types of knowledge (i.e., landmark, route, survey), spatial relations form the *building blocks* of environmental representations. They appear to be processed by at least partially distinct neuronal pathways in the left and right hemisphere, which encode categorical and coordinate spatial relations, respectively (e.g., [16,17]). The relationship between reference frame processing and spatial relation coding has been investigated (e.g., [18]) showing how, notwithstanding categorical judgements being more accurate than coordinate ones, with both egocentric and allocentric reference frames, the difference between the two types of spatial relations is stronger when combined with allocentric information.

### 1.2. Spatial Mental Representations, Aging, and Familiarity with the Environment

With the maturation of brain regions implicated in spatial reasoning and continual interaction with their environments from childhood to adulthood, people become increasingly competent in their use of topographical information (e.g., [19,20]). At the other end of the spectrum, even this well-developed skill is destined to decline with age. Numerous cognitive and behavioral changes occur with aging, typically including pervasive declines in, among others, memory, attention, perceptual and executive capacity. Importantly, aging has also been associated with functional decline in spatial cognition on a spectrum which spans normal aging to mild cognitive impairment to dementia (e.g., [21]). With regard to spatial memory, significant age-related decline has been observed in episodic spatial memory and spatial learning, spatial working memory, and egocentric and allocentric memory for spatial locations (for a review, [22]). In the elderly, episodic spatial memory impairment is an early clinical manifestation of cognitive decline associated with difficulty learning new spatial information (for a review, see [23]) and mainly affects allocentric rather than egocentric spatial cognition (e.g., [21,24,25,26,27]). The effect of age on visuospatial perception has been studied in relation to categorical and coordinate spatial relations. It appears that healthy elderly populations can judge global positions more easily than they can estimate exact distances, consolidating the notion that coordinate processing may be more affected by aging (e.g., [28,29,30]). These behavioral findings are supported by age-related loss and impairments in several brain circuits (e.g., [31,32,33,34,35]).

It is known that the ability to keep spatial information in memory decreases with age, but there is firm evidence that well-consolidated spatial memories are less prone to impairment or loss than recently learned memories [36,37]. It is likely that normal changes in brain structure and function make it slightly harder to learn new information quickly (e.g., [38]) and to filter out distractions that can interfere with that learning (e.g., [39]).

Notwithstanding these findings, the robustness and durability of remote memory traces does not depend solely on the time that spatial information has been stored in memory’ (e.g., [40,41]). Extended experience can also counter the erosion of memory traces. Experience may act as a protective factor, since practice within an environment enhances the consolidation of spatial memory traces (e.g., [42]). Remote memory traces that are not refreshed over time may degrade more slowly than recently learned memories, but they are not immune to being forgotten. By contrast, remote memory traces that are repeatedly reinforced suffer only marginal effects of forgetting. Hence, in aging, remote memory traces refreshed over time are resistant to forgetting, remote memories that are not repeated over time and recurrent recent memory traces decay more quickly, and finally, recent memories that are not repeated do not even get to the process of memory consolidation (e.g., [43]).

The frequent experience of environmental exploration implicitly entailed by activities of daily living, such as doing the shopping, going for a walk, going to the doctor, and working, might be important for preserving spatial memory function and play an important role in maintaining spatial cognition [44,45]. In conclusion, here we support the notion of familiarity as related to a combination of trace remoteness and frequency of re-exposure at the level of experience (e.g., [14,23,46]).

Familiarity has previously been conceptualized as the amount of time spent in a place, how much attention a person has focused on the place, how well a person thinks they know the place, or how well a person knows the place according to an objective test. Less importance has been attributed to the format of spatial information acquisition, for instance, map study vs. navigation. Indeed, Evans and Pezdek [47] tested young participants on a university campus in San Bernardino (CA, USA) with which they were *familiar* or *unfamiliar.* Unfamiliar participants had learned the campus environment through map study, whereas familiar participants relied on their long-term direct experience of navigating the campus. Participants had to decide whether sets of three buildings were arranged in the correct spatial configuration (incorrect sets were mirror images of the correct layout). An effect of orientation was found only when buildings were unfamiliar (difficulty recognizing unfamiliar rotated stimuli was found to be a linear function of angular rotation). The results suggested that orientation-dependent representations (with respect to experienced views) were produced from map study, while orientation-independent representations (based on experience of the environment from several viewpoints) emerged when spatial information was acquired directly from the real environment. Soon after, Thorndyke and Hayes-Roth [9] found that direct experience with the environment allowed familiar participants to perform better on tasks related to route estimation and pointing out objects than the unfamiliar group, who had learned the same spatial information through map study.

Notwithstanding the importance of these studies, they suffer from a methodological problem: the familiar and unfamiliar group were not fully comparable, because each experimental condition mixed two sources of variation: the format of spatial information acquisition and the duration of exposure to that information.

Further research focused on the use of environmental information acquired by navigating places in which participants had resided for a long time. Rosenbaum and colleagues [40] showed that elderly people performed as well as the young on a judgment of direction task (practically speaking, a categorical spatial relation task) situated in the city of Toronto, where they had lived for at least 10 years but had returned only rarely recently. By contrast, their performance was worse than the young’s in relation to a new environment, learned through virtual navigation. Meneghetti and colleagues [48] confirmed that both young and older people performed with the same level of accuracy on judgment of direction tasks related to the city of Reggio Emilia, Italy. Campbell and colleagues [49] showed that aging had no impact on memory for a familiar environment (downtown Sydney, Australia), in an ecological task involving a topographical map and landmark assessment. Muffato and colleagues [50] used a pointing task, asking participants to imagine pointing in the direction of a given landmark in the Botanical Garden in Padua while standing next to another landmark and facing a third. They showed that the spatial mental representations of elderly people were as accurate as those of the young, whereas the former performed worse than the latter after studying a map of a new environment. Merriman and colleagues [51], using object recognition, egocentric direction judgement, and allocentric proximity judgement tasks based on areas within Trinity College, Dublin, suggested that familiarity was a facilitating factor in aging only when spatial memory relied on an egocentric frame. Lopez and colleagues [52] suggested that repeated exploration of a hometown allowed for the consolidation of memory traces, delaying their deterioration with aging. They used two different hometown tasks related to the city of Bari (adopting an egocentric and an allocentric frame of reference, respectively) to test categorical spatial relations. Moreover, in a parallel study, the same authors [53] observed no age effects on either task, whereas such effects were present in a task based on more recently acquired spatial information. Muffato and Meneghetti [54] using pointing, landmark-locating, and shortest-path-finding tasks, showed that young participants with high familiarity with the university campus of Padua were better able to identify the shortest way to a destination, compared to a group with low familiarity. No differences emerged between the two groups on the other tasks. Caffò and colleagues [55], employing the same hometown tasks as Lopez and colleagues [53], showed that familiarity (spatial memory traces) can be a protective factor for retrospective components of topographical disorientation in normal aging. Indeed, egocentric and allocentric components linked to the recall of familiar spatial knowledge were relatively spared with respect to learning unfamiliar (new) knowledge in normal aging, but not in elderly people with Mild Cognitive Impairment. Lastly, Lopez and colleagues [56], using sketch map tasks and testing categorical and coordinate spatial relations, showed that the young and the elderly performed equally well on the hometown task related to the city of Turi. There was a reduction in the difference between coordinate and categorical accuracy with increasing familiarity with the environment.

This brief review makes it clear that familiarity is a key concept in spatial cognition studies, although it has not been explicitly identified as such by researchers. At first, familiarity was thought to reflect a mixture of time spent in a place using a specific mode of information acquisition, and knowledge according to an objective test [9,47]. More recently, the concept has been operationalized as the amount of time participants have spent in their hometown and how well they think they know a place [40,48,49,50,52,53,55]. According to the *Environmental Knowledge Model* (EKM, [57]) familiarity can be considered the main factor required to predict the ability to solve environmental tasks, regardless of spatial abilities. In the present study, the notion of familiarity is understood in terms of potential and actual exposure to/interaction with the hometown environment.

### 1.3. An Extended Way of Conceptualizing Familiarity: the Environmental Affordances of a Place

The environment contains clues that indicate possibilities for action. Gibson [58] wrote that: “The ‘values’ and ‘meanings’ of things in the environment can be directly perceived” (p. 127), in the sense that “postboxes afford letter mailing” (p. 139). Gibson [59] coined the term “affordance” to indicate an opportunity the environment offers an animal given that animal’s capacity for action (for a review, [60]). Borrowing this model provides new insight into human spatial cognition. Dominant elements in the environment such as landmarks, districts, nodes, paths, and edges [61] can be considered spatial affordances for human beings, because these features openly declare their functions, *as possibilities for action*. Landmarks are points of reference, districts and nodes signal discrete areas, paths indicate turns, and edges suggest crossings. As observed by Norman [62], every environmental element per se is characterized by a *physical affordance*.

Moreover, everyone can assign meanings to and provide information on the conventional uses of familiar tools, based on their personal experience with them. This second aspect corresponds to the notion of a *learned affordance*, in the sense of the personal meaning [62] something acquires. Returning to the topic of spatial cognition, consider the example of a bridge. *Brooklyn Bridge* in New York has completely different meanings and invokes different potential actions for a native New Yorker and a tourist, because of their different levels of experience with the city. As Pezzulo and Cisek [63] have noted with reference to ecological contexts, affordances emerge from interactions between human beings and their environment: the environment *itself* becomes more affordable for humans because of these interactions. Hence, if familiarity with an environment is nothing more than interactions experienced within it, and higher levels of familiarity correspond to a larger number of previous interactions, an individual’s *level of personal experience with that environment* can be considered a kind of *learned affordance*.

Finally, as Ramenzoni and colleagues [64] point out, humans can perceive affordances not only for themselves, but also for others. They are able to distinguish their own action capabilities from those of others [65,66], by using their own embodied simulation models (for more details, see [67]). In the spatial domain, the implication is that being an *active* viewer of the environment might improve the quality of one’s affordances compared to a *passive* viewer, who does not act directly but only sees another agent in action [68]. In the realm of spatial cognition, this eventuality often occurs, for instance, when we are a passenger in a car driven by someone else or on a bus. Whether one explores the environment in an *active or passive mode* could mediate the learned affordances, and consequently our sense of familiarity with a place.

### 1.4. Factors that Influence the Relation between Familiarity and Spatial Mental Representations

Several factors can play a role in the relation between familiarity and spatial mental representations (for a review, see [69]). These include residential zones, socio-economic status, gender, education, and cognitive efficiency/intelligence. Incomes and residential locations are widely considered relevant since it has been found that people with higher incomes are more likely to fully benefit from their living environments (e.g., [70,71]). They tend to be more engaged in outdoor activities that involve movements and actions beyond those required for subsistence. Moreover, higher levels of active navigation appear to be associated with the places people reside: living in a suburb, away from the main activities of the city, gives inhabitants less opportunity to interact with the environment (e.g., [72,73]). As for gender, environment-related differences in behavior have been reported, with respect to income, car ownership, participation in the workforce, education, and choice of residence (e.g., [74]). Gender may also be a factor in the quality of spatial mental representations: males are reportedly better at distance estimations, while women surpass men in making categorical judgments (e.g., [75,76]). As for education and cognitive functioning, different levels of schooling and cognitive abilities influence everyday spatial functional abilities (e.g., [55,77]). Especially in elderly people, a higher level of education and cognitive functioning increases their engagement with their environment, thereby allowing them to maintain spatial information longer (e.g., [53,78] presumably by increasing cognitive reserve [79]). Therefore, education and cognitive functioning are associated with adult socio-economic status and active social behavior, promoting an active lifestyle through physical activity and active navigation in the environment [80,81,82,83]. Moreover, both level of education and cognitive abilities influence allocentric topographic knowledge (e.g., [55,56,84]).

### 1.5. The Present Study

In light of the above discussion, we can conclude that residential zones, socio-economic status, gender, education, and cognitive efficiency/intelligence can be used as covariates of spatial performance. It is assumed that age could moderate the relationship between the indices of experience and performance in the mental representation of familiar environments. In other words, the relationship between experience and spatial mental representations is expected to assume different weightings for different age groups.

For the first time, the present study brings together the abovementioned variables, systematically mapping their contributions to different aspects of spatial performance. The starting point is Figure 1, which shows the overall theoretical moderated model. This model represents the general hypothesis that the effect of familiarity (understood as exploration of the environment) on spatial relations, is moderated by age, while controlling for some factors that could influence that relationship. This general theoretical model was applied to test four specific moderated models by using frequency and mode of exploration as predictors, and categorical and coordinate spatial relations as outcomes. The hypothesis tested is that a high frequency and active modes of exploration have positive effects on spatial mental representations and intersect with the age of participants. More time spent exploring the hometown should allow for a more consolidated memory trace and, in turn, a better performance with spatial mental representations. Moreover, based on previous studies, no significant effect of age was expected, except on categorical spatial relations.

## 2. Materials and Methods

### 2.1. Participants

A power analysis to estimate the required sample size was carried out using G*Power 3.1 [85]. The sample size was established by considering three aspects: a level of significance equal to 0.05, a power of 0.80, and the effect size. The analysis indicated that there was an 80% chance of correctly rejecting the null hypothesis of no effect given a total of 759 participants for a cautious low effect size (0.02), 109 for a medium effect size (0.15) and 52 for a large effect size (0.35). Given that there was no similar study in the literature to use as a basis to hypothesize the effect size, a small effect size should have been assumed as a conservative approach. However, it was already challenging to collect enough data for a medium effect size; therefore, data were collected to surpass the estimated sample for a medium effect size (i.e., N ≅ 100). Approximately four times the estimated sample associated with a medium effect size was considered an adequate sample dimension (e.g., [86]).

At the end of the enrolment procedure, four hundred healthy participants (201 women) took part in the study. All participants were from the metropolitan area of Bari, Apulia, Italy. The city of Bari has a population of 320,257 inhabitants, over 116 square kilometers (45 sq mi), while the urban area has 750,000 inhabitants. The metropolitan area has 1.3 million inhabitants. Two hundred and three young university students (i.e., age mean ± sd 23.61 ± 5.53; level of education mean ± sd 16.17 ± 1.61), and 197 Elderly people (i.e., age mean ± sd 73.06 ± 6.80; level of education mean ± sd 11.47 ± 5.08) were enrolled in the study. Descriptive statistics for the two groups are reported in Table 1. All participants, blind to the hypothesis of the study, signed a consensus form. The local ethical committee (No. 3660-CEL03/17 November 2017) approved the study protocol, and the whole study was performed following the Helsinki Declaration and its later amendments.

The inclusion criteria for all participants were: (a) having lived in Bari from birth, and (b) being able to recall the names of the eight landmarks displayed on the 21 × 15 cm photographs, used in previous research [46,52,53], and (c) normal cognitive function.

### 2.2. Procedure and Materials

Participants were assessed individually in a well-lit, quiet room without disturbances. Each step in the testing procedure was made clear to the participants beforehand. Data were collected in one session ranging between 60 and 90 min for the elderly, and 30 and 60 min for the young. Breaks were allowed on request.

Elderly participants were volunteers recruited from senior centres and third-age universities with the support of an informant, generally an undergraduate or graduate student, trainees, or employees of the centres. They were instructed to reach out to people known to be in a good state of general physical and psychological health. Elderly participants were consecutively enrolled between February 2018 and July 2019. First, in order to exclude people with a history of suspected uncompensated systemic/traumatic/psychiatric disease, or with severe vision/hearing loss, which could have affected cognition, a general anamnesis was assessed by supervised trainees in psychogeriatric assessment. Furthermore, global cognitive function was assessed by the Montreal Cognitive Assessment (MoCA, [87,88,89,90]), with an inclusion cut-off above 17. Forty-eight participants were excluded, because they did not meet the inclusion criterion. Moreover, the Activities of Daily Living and Instrumental Activities of Daily Living (ADL, [53,91]; IADL, [53,92], inclusion cut-off higher than four for ADL, women: mean ± sd 5.98 ± 0.19, men: 5.83 ± 0.48; and higher than four for males and six for females for IADL, women: mean ± sd 6.54 ± 1.50; men: 5.58 ± 2.00), the 15-item version of the Geriatric Depression Scale (GDS, [53,93]; inclusion cut-off less than four, women: mean ± sd 2.10 ± 2.08; men: 3.20 ± 2.80), and the Subjective Memory Complaints Questionnaire (SMCQ, [53,94], inclusion cut-off less than five, women: mean ± sd 2.34 ± 1.40; men: 2.80 ± 1.40) were administered. This did not give any further exclusions. Elderly females were difficult to recruit and had to be intentionally sought out. All other subgroups were easily filled. Data from excluded participants were not recorded. Young participants were recruited from the University of Bari and they were at the first year of a master’s degree in Psychology. They were enrolled between February 2018 and July 2019. A general anamnesis and the MoCA test were administered to them by supervised trainees. At the end of the enrolment procedure, the final sample was composed of 197 elderly and 203 young participants.

Moreover, two other variables related to the location of their home were ascertained from participant report: (a) their residential location based on proximity to the downtown area, following the classical Burgess’ Concentric Zone Model [95]. The first zone comprised the administrative core of the city and the city center (central district, labelled “1”); the second was an area largely comprising housing and special commodities (transition zone, labelled “2”), and the last zone was the periphery of the city (suburb, labelled “3”); and (b) a measure of socio-economic status derived from rental costs or property values in their neighborhood (e.g., [96].), depending on its location within the central district, transition zone, or suburb. Finally, the Level of Experience Index (LEX), the Passive Active Transportation index (PAT), and Landmark positioning on a map (LPM) task were assessed. The last three measures will be described below.

### 2.3. Measures of Exploration

All participants were evaluated on their level of experience with their hometown through a structured interview. Two composite measures were obtained: the Level of Experience Index (LEX), and the Passive Active Transportation index (PAT). Both indexes are included in an index of activity “in daily routines” called the Hometown Index of Exposure [53]. The Hometown Index of Exposure rated participants as active if they responded that they moved around within their hometown by foot and/or using vehicles at least three times a week, combining information about frequency and mode of transportation. In the light of a better specification for the component of familiarity with the hometown environment, all the participants in this study were rated as low or high on LEX, based on how many times a week they moved around their hometown—no more than two times (low) or more than two times (high)—to shop, go for a walk, attend mass, visit to the doctor, play sports, work/study, and as a leisure activity [97]. Moreover, for each category listed above, all participants were rated as mainly passive or active navigators, using the Passive Active Transportation index (PAT), a measure of modal experience inspired by Mondschein and colleagues [98]. Active modes of transportation include driving a car, scooter or bike, and walking, which require people to make wayfinding decisions during their navigation. Likewise, the use of public transit and being a passenger in a car are considered passive modes because they require no ongoing purposeful wayfinding ability (e.g., [6,99]). Participants were classified as passive or active navigators if most of their weekly explorations were associated with an active or passive mode of transportation, respectively. Means and standard deviations for each test and for each group of participants are reported in Table 1.

### 2.4. Hometown Task: Landmark Positioning on a Map (LPM)

Afterwards, participants were required to complete landmark positioning on a map task (LPM, [46]). Participants were first required to recognize 10 well-known landmarks in their hometown that were displayed in photographs (Figure 2a,b). The given references for the hometown map were the marking for North, and two of the 10 landmarks, namely one in the centre of a semi-blind map and the other further outside the city on the map (Figure 2c). The two landmarks served as positional and distance reference points that allowed participants to infer positions and distances between the other landmarks. The participant had to pinpoint all the other eight landmarks, keeping in mind the metric (i.e., relative distances) as well as categorical (relative positions) spatial relations between landmarks (Figure 2d). The LPM task is known to be relatively unbiased with respect to age and education, and relatively sensitive in discriminating between different levels of cognitive functioning [53].

### 2.5. Scoring Method

In order to compute the scores of categorical and coordinate spatial relations, the scoring method applied by Lopez and colleagues [100,101] to disentangle categorical and coordinate information on three landmarks located in a sketching area, was adopted for the LPM task. Using custom-made Excel Macros, both categorical and coordinate information was gathered. Categorical judgements were assessed for each pair of landmarks separately on the x (e.g., G is to the left of D) and y axes (e.g., D is above G). For each correct categorical spatial judgement, participants were awarded from 1 to a maximum of 56 points (i.e., 28 points for each axis). To assess coordinate judgements, we considered the axial components of Manhattan distance. Coordinate judgments were made for each pair of landmarks by comparing distances, separately, on the x axis (e.g., the distance between landmarks G and D is greater than the distance between landmarks F and B) and y axis (e.g., the distance between landmarks G and D is less than the distance between F and B). For each correct coordinate spatial relation, participants were awarded from 1 to a maximum of 756 points (i.e., 378 points for each axis). The final scores for categorical and coordinate spatial relations were transformed into the proportion of correct responses (range between 0 and 1, e.g., [102]) The reliability of the task based on categorical and coordinate spatial relations was also computed. Cronbach’s Alpha scores were 0.91 and 0.85 for categories on the x and y axis, respectively, and 0.88 and 0.91 for coordinates on the x and y axis respectively.

### 2.6. Statistical Analysis

Descriptive statistics, preliminary analyses of neuropsychological tests, and measures of familiarity with the environment were performed. In order to test gender and age group differences for demographical variables, spatial relations and levels of familiarity with environment, χ^2^ (for gender), Analysis of Variance (for age, education, income, neuropsychological tests, spatial relations) and the Cochran–Mantel–Haenszel test (for residential location and measures of familiarity with environment) were performed. Furthermore, with the purpose of verifying if subsequent analyses could be carried out separately for the two components of familiarity (e.g., LEX and PAT), a Multivariate Analysis of Variance was run with categorical and coordinate spatial relations as the continuous response variable and LEX × PAT as a single factor variable. The aim of this preliminary analysis was to exclude an interaction effect between the two variables constituting one of the foci of the present study. Finally, in order to achieve the aims of the present study, a SPSS macro-Process, Model 1-developed by Preacher and Hayes [103] was used to test four moderated models, namely the effect of familiarity in terms of experience through the use of LEX (models 1, 3) and PAT (models 2, 4), on performance in terms of categorical (models 1, 2) and coordinate (models 3, 4) relations. In order to test if the effect of familiarity on performance was moderated by age, and if there were age differences in performance, the variable Age Group was put in the model as a moderator, with the aim of testing its influence on the hypothetical relation between the intended measure of familiarity and the outcome. Moderation occurs when the effect of familiarity is different for different levels of Age Group (i.e., levels of the moderator). Gender, education, cognitive functioning, income, and proximity to the city centre were controlled as covariates (see Figure 1). The effect size of slope coefficients was computed by means of Cohen’s *d*, with coefficients of 0.20, 0.50, 0.80 representing roughly small, medium, and large effect sizes, respectively [104].

## 3. Results

Descriptive statistics and preliminary analyses on neuropsychological tests and measures of familiarity with the environment are reported in Table 1. Young and older participants showed significant differences in their level of education and cognitive functioning in favour of the young. No differences in residential location and income were reported for young and elderly participants. As for levels of exploration of the environment, the elderly reported, on average, fewer interactions with their environment (LEX) than the young, but there were no differences in the use of different modes of transportation (PAT). The young outperformed the elderly in terms of coordinate, while there was a substantial balance on categorical spatial relations. As for gender differences, these emerged only in older participants: older women performed worse than men on both categorical and coordinate spatial relations (see also Appendix A).

The MANOVA found no significant interaction effect of LEX × PAT on categorical and coordinate spatial relations (Wilks’ λ = 1.00, F(2395) = 0.41, *p* = 0.66). This legitimized their separation as factors in further analyses.

In order to satisfy the aims of the present study, four moderated models were performed. The results were as follows.

### 3.1. Model 1. Level of Experience (LEX) on Categorical Spatial Relations

The total model was significant (F(8391) = 31.40, *p* < 0.001) and explained about 39% of the variance in performance on categorical spatial relations. In this model, the interaction LEX × Age Group significantly increased the R-squared of 2% (F = 11.46, *p* < 0.001). Table 2 shows all the effects on categorical spatial relations. Controlling for covariates (i.e., gender, education, cognitive functioning, income, and proximity to the city center), the effects of LEX, Group Age, as well as their interaction, on categorical judgments, were all significant. Overall, the higher the familiarity, the higher the performance. Young participants outperformed the elderly. More importantly, the conditional effect of LEX on categorical judgments at the value of the moderator showed a stronger effect in the elderly. High levels of LEX doubled the performance of the elderly compared to that of the young. Consequently, the positive effect of LEX on outcomes was twice as high for the elderly as for young participants (see also Appendix A).

### 3.2. Model 2. Passive Active Transportation (PAT) on Categorical Spatial Relations

The total model was significant (F(8391) = 8.45, *p* < 0.001) and explained about 15% of variance in categorical spatial relations. Considering the interaction PAT × Group Age, it significantly increased the explained variance in performance (R-squared = 2%, F = 6.00, *p* = 0.014). Table 3 shows all the effects on categorical spatial relations. Controlling for covariates, the principal effects of PAT as well as Group Age on categorial spatial relations were not significant, while their interaction was significant (see also Appendix A). Indeed, being an active explorer significantly predicted performance only among the elderly. Finally, education also seemed to exert a positive effect on performance in the present model.

### 3.3. Model 3. Level of Experience (LEX) on Coordinate Spatial Relations

The total model was significant (F(8391) = 39.89, *p* < 0.001) and explained about 45% of the variance in coordinate spatial relations, with a significant contribution from the interaction LEX × Group Age, increasing R-squared by approximately 3% (F = 17.78, *p* < 0.001). Table 4 shows all the effects on coordinate spatial relations. Controlling for covariates, the effects of LEX, Group Age, as well as their interaction, on coordinate judgments, were significant. Overall, the higher the experience, the higher the performance. Young participants outdid the elderly in sketching coordinate information. In this case, the conditional effect of LEX also showed a stronger effect in the elderly. High levels of LEX doubled the performance of the elderly compared to that of the young. Consequently, the beneficial effect of LEX on the elderly was double that on young participants (see also Appendix A).

### 3.4. Model 4. Passive Active Transportation (PAT) on Coordinate Spatial Relations

The total model was significant (F(8391) = 13.79, *p* < 0.001) and explained about 22% of variance in coordinate spatial relations. In this model, the interaction PAT × Group Age increased R-squared by approximately 1% (F = 3.91, *p* = 0.038). Table 5 shows all the effects on coordinate spatial relations. Controlling for covariates, using a passive or active mode of transportation had no significant effect on coordinate performance. Instead, Group Age and its interaction with PAT were significant (see also Appendix A). Only the elderly gained an advantage when required to estimate distances between very familiar landmarks in their hometown by being active navigators in their hometown. Finally, cognitive functioning and education seemed to exert a positive effect on performance.

Overall, both LEX and PAT explained more variance in performance with coordinate than categorical spatial relations. Moreover, LEX explained more variance than PAT, in both categorical and coordinate spatial relations.

## 4. Discussion

A recent study (e.g., [105]) highlighted the importance of evaluating spatial cognition in aging with respect to a well-known daily living environment (i.e., hometown). The present study aimed to evaluate the moderating effect of age on the relation between familiarity with environment (i.e., level of experience and passive/active transportation) and spatial mental representations of the hometown in young and elderly people who had lived there since birth.

The literature examining familiarity with the environment has, to date, broadly conceived of familiarity as *the level of knowledge* with an environment measured as the time spent in a place (e.g., [54]). As stated in the introduction section of the present study, familiarity with the environment could be also considered an extension of the concept of affordance: the way in which people approach/interact with their environments is guided by different purposes (e.g., [106,107]). Landmarks carry with them general meanings and functions that make them particularly helpful for navigating new environments (“that looks like *a hospital*”, “I see *a bridge* ahead that crosses the river”) [108]. Further, their *individual* meanings become particularly helpful in well-known environments (“that building is *the Municipality, where I requested the identification document*”, “this is *the right turn that takes us home*”) (e.g., [63]).

The two groups of participants enrolled in the present study consisted of healthy people with normal cognitive functioning who exhibited differences in terms of frequency of exploration. The young declared that they moved around their hometown more often than the elderly. This evidence is in line with the literature, which underscores that the young are more likely to move around on account of their jobs, education, physical and leisure activities (e.g., [109]). However, the groups did not differ in terms of their use of different modes of transportation. In this respect, our results converge again with previous studies (e.g., [110]), which compared transport choices in the young and elderly, showing no substantial differences. Overall, the main findings suggested that there were moderate age differences in categorical spatial relations (in the female sub-sample), larger age differences in coordinate spatial relations, and both measures of exploration had different effects on performance with categorical and coordinate spatial relations in the young compared to the elderly participants.

Considering the moderated models, with respect to the effect of age on categorical and coordinate spatial relations, young and elderly participants showed a non-significant (PAT) and a significant but small difference (LEX) in categorical performance. This result is partially in line with previous findings [52,53]. However, in the case of coordinate spatial relations, the two groups of participants showed a larger difference in their sketches when both PAT and LEX were considered in the moderated models. This evidence extended to remote spatial information, thereby confirming the results of Bruyer and colleagues [34], who reported a decrease in performance on coordinate spatial judgements with aging, when using a new set of information.

As for the role of familiarity, a higher frequency of exploration positively affected the representation of the environment, both in terms of categories and coordinates. Moreover, age moderated that relation. In fact, experience with the hometown impacted the performance of the elderly more than the young. These findings were independent of estimated income, residential location, gender, education, and cognitive functioning, all factors which are known to be influential (e.g., [70,72,73]). In other words, both the young and the elderly took advantage of direct exploration and interaction with the environment (level of experience); however, the elderly benefited almost twice as much as the young participants in both categorical and coordinate judgments. Moreover, the active mode of exploration positively predicted higher performance in the case of both categorical and coordinate spatial relations in the elderly. In other words, having more experience with the environment, *and* being frequently/actively exposed to this environment, increased the possibility of maintaining accurate spatial mental representations of their hometown over time for older participants. Hence, being active navigators of their hometown, as drivers or walkers, allowed the elderly to update their route position with respect to landmarks present in the environment. Ongoing wayfinding helped them maintain exact spatial mental images, enhancing their allocentric spatial mental representations (e.g., [111]). By contrast, the young did not need to be active navigators to effectively represent the environment. Because young people already enjoy the highest level of cognitive efficiency (e.g., [112]), their cognitive mapping is stronger than that of older people [113].

In conclusion, maintaining a high frequency of activity in the environment acts as cognitive training, contributing to the plasticity of cognitive functioning (e.g., [114]). Repeated exploration of the environment helps the elderly preserve and update their sense of direction and estimation of distances, and their ability to construct flexible spatial mental representations. In sum, the young profit from their cognitive abilities, while the elderly benefit from familiarity.

Furthermore, as a non-trivial side-effect, activity in aging has been associated with good general health and well-being (e.g., [115]). In particular, feeling stimulated by being active enhances the motivation of older adults and promotes their physical and mental health [116]. From the perspective of active aging, it is important to know that continuing to encounter, explore, and experience one’s habitat slows memory deterioration and loss and represents a benefit for the elderly, in line with the European policy of supporting the *emergence of successful aging* (e.g., [117]).

This study had some limitations. First, familiarity was evaluated using a self-report questionnaire, which can be biased by positive/negative tendencies not controlled in the present study. Moreover, familiarity as the frequency and mode (active/passive) of exploration could have been measured by using GPS/datalogger data (e.g., [118]). Furthermore, it will be necessary to arrange a series of experiments using a computer-administered version of the LPM task, composed of landmarks that are particularly salient for participants in relation to their personal experience. Last, interaction with an environment does not completely cover the concept of familiarity with the environment, because it does not take into account how many times people frequent a specific landmark or the subjective attribution which each individual assigns to spatial encounters, namely, their personal sense of familiarity with an environment. According to Kaplan and Kaplan [119] (p. 92), familiarity is: “*The relationship between an individual and something that individual has had considerable experience with […]. Familiarity thus provides the ways to facilitate the basic process of searching for a solution and connecting startup and goal*” (pp. 166–167).

This is a sort of feeling of confidence with places or landmarks developed on the basis of repeated and frequent exposure to and interactions with an environment (e.g., [120,121]). Future research will need to measure the personal sense of familiarity of the participants as well.

Despite these limitations, the current research has several strengths and introduces some valuable innovations. The scoring method used to disentangle categorical and coordinate spatial relations [100,101] has been applied to a simplified version of sketch maps composed of eight landmarks (LPM, [46]). This demonstrates the suitability of a method that is easy and quick to apply to maps characterized by the presence of labelled landmarks placed in a Cartesian plane. The next goal would be to develop software that can automatically code measures for the distinct analysis of positions and distances between points in a geographic space. Moreover, the method of separating coordinate and categorical components in an integrated external representation of spatial information is useful for independently decoding position and distance. Especially in aging research, knowing the trajectories of categorical and coordinate spatial relations, the building blocks of spatial mental representations, can help the researcher better understand the online processing of navigation and offline spatial reasoning, and the functioning of their neural correlates (e.g., [122,123]).

Finally, it has been shown, once again, that using an ecologically valid task based on participants’ hometown knowledge is a suitable method for monitoring topographical orientation in aging. It also highlights the role of frequent and active exploration of the environment in aging in maintaining functional independence in daily living activities.

## 5. Conclusions

“After crosses and losses men grow humbler and wiser”. Borrowing Benjamin Franklin’s aphorism and applying it to spatial cognition, we can state that people become more confident with their environment after a wealth of experiences, which in turn enriches their spatial knowledge. The wisdom of the elderly lies in the fact that older people benefit more from experience with their hometown than young people.

Familiarity, conceived in terms of frequency and active/passive modes of exploration, seems to be a particularly promising approach to evaluating spatial mental representations, and promoting specific interventions and contributing to the construction of preventive and rehabilitative interventions in aging (e.g., [124]). Moreover, navigation frequency and active modes of transport are more highly associated with better performance in the elderly than the young. Young people are likely to take advantage of their efficient cognitive system, and are therefore less sensitive to the positive effects of frequent exploration and high levels of activity/independence when navigating. An active experience of the environment might contribute to environmental cognitive reserve in the elderly (e.g., [79,125,126]). Finally, the present research contributes by underscoring earlier findings, in which the importance of evaluating spatial mental representations in daily living environments has emerged. By being active navigators who intensively and continually explore their environments, older adults can maintain more accurate spatial representations and orientation skills and improve their general health. In other words, while being less active in the environment could have trivial consequences for a young person, it has far greater consequences for an older person, who can benefit from the strong relation between frequent/active exploration of the environment, spatial mental representations and, in turn, orientation skills. In conclusion, extensive and purposeful experience with the environment is an important ecological source of cognitive stimulation, which significantly supports physical, psychological, and cognitive well-being during aging.

## Figures and Tables

**Figure 1 ijerph-18-00709-f001:**
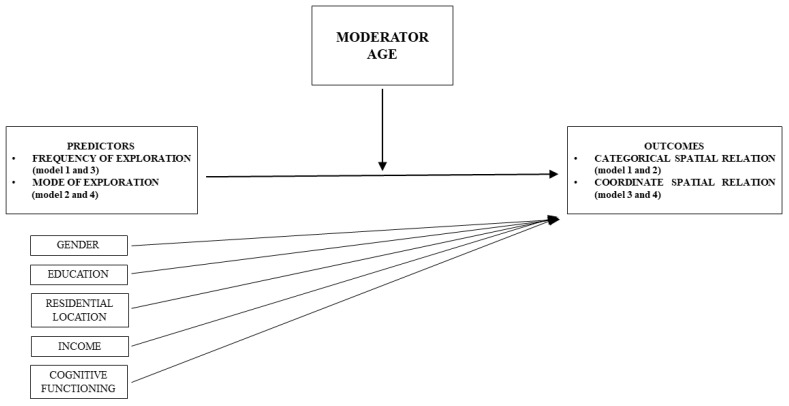
The overall theoretical moderated model.

**Figure 2 ijerph-18-00709-f002:**
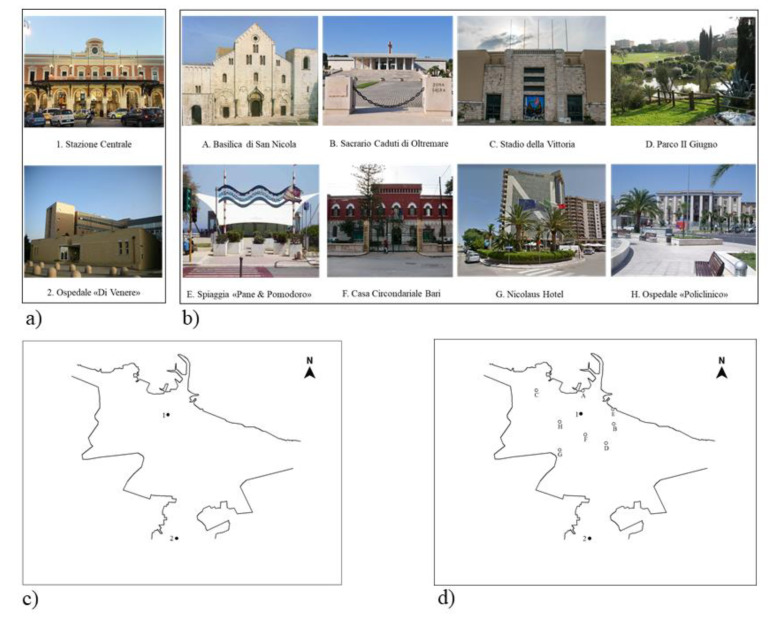
Map of Hometown, City of Bari: (**a**) pictures of fixed reference points; (**b**) pictures of landmarks to be placed on the semi-blind map; (**c**) semi-blind map with two fixed reference points and North; (**d**) scoring sheet: the map with the expected positions and distances. Every correct landmark placement corresponds to a correct comparison between landmark position and side (the landmark position: above and below x-Cartesian coordinate; the landmark side: right and left y-Cartesian coordinate).

**Table 1 ijerph-18-00709-t001:** Means ± standard deviations for intervals and frequencies for categorical variables. Significant main effects of age-group and gender on demographical variables, spatial relations, and levels of familiarity with the environment, were obtained through *t* test −χ2 for frequencies. To test the interaction of Age Group × Gender, we used the Cochran–Mantel–Haenszel test for stratified categorical data and conducted a series of ANOVA(s) between subjects. * *p* < 0.05, ** *p* < 0.01, *** *p* < 0.001.

	Young		Elderly				
	Women	Men	Women	Men			
	(N = 103)	(N = 100)	(N = 98)	(N = 99)			
					Group Age	Gender	Interaction
Age, years	24 ± 6	23 ± 6	73 ± 7	73 ± 7	*****	*-*	*-*
Education, years	16 ± 2	16 ± 1	11 ± 5	12 ± 5	*****	*n.s*	*n.s*
Residential Location (Centre, Transition, Suburbs)	86/5/12	80/2/18	71/3/24	82/1/16	*n.s.*	*n.s.*	*n.s.*
Income	60 ± 11	58 ± 11	55 ± 9	56 ± 9	*n.s.*	*n.s.*	*n.s.*
Montreal Cognitive Assessment	28.5 ± 1.6	29.5 ± 0.5	23.2 ± 4.1	23.6 ± 3.9	*****	*n.s.*	*n.s.*
FAMILIARITY WITH ENVIRONMENT							
Low/High Level of Experience	21/82	64/36	66/32	45/54	***	*n.s.*	*n.s.*
Passive/Active Trasportation	48/55	51/49	50/48	55/44	*n.s.*	*n.s.*	*n.s.*
SPATIAL RELATIONS							
Category	0.76 ± 0.11	0.74 ± 0.24	0.66 ± 0.18	0.72 ± 0.20	*n.s.*	*n.s.*	*****
Coordinate	0.75 ± 0.10	0.73 ± 0.16	0.60 ± 0.12	0.67 ± 0.14	****	*n.s.*	****

**Table 2 ijerph-18-00709-t002:** Standardized coefficients of the moderated Model 1.

	MODEL 1				
	β	Std.Err	t	*p*-Value	Cohen’s d
Total sample (N = 400)					
Lex→Cat	1.053	0.084	12.52	<0.001	0.57
Group Age→Cat	0.305	0.076	2.77	0.006	0.15
MoCA→Cat	0.013	0.014	0.905	0.350	-
Education→Cat	0.004	0.011	0.389	0.707	-
Gender→Cat	−0.066	0.042	−1.56	0.121	-
Proximity→Cat	0.090	0.061	1.47	0.120	-
Income→Cat	0.001	0.004	0.321	0.750	-
Lex * Group Age→Cat	−0.310	0.087	−3.38	<0.001	0.17
Young (N = 203)					
Lex→Cat	0.750	0.11	6.34	<0.001	0.40
Elderly (N = 197)					
Lex→Cat	1.400	0.12	11.11	<0.001	0.70

Note: Lex = Level of Experience, Cat = Categorical Spatial Relations, Group Age = Elderly vs. Young, MoCA = Montreal Cognitive Assessment, Education = Level of Education, Gender = Male and Female, Proximity = Residential Location, Income = Socio-Economic Status, * means interaction between reported variables.

**Table 3 ijerph-18-00709-t003:** Standardized coefficients of the moderated Model 2.

	MODEL 2				
	β	Std.Err	t	*p*-Value	Cohen’s d
Total sample (N = 400)					
Pat→Cat	0.103	0.094	1.00	0.323	-
Group Age→Cat	0.206	0.080	1.36	0.172	-
MoCA→Cat	0.027	0.017	1.58	0.114	-
Education→Cat	0.047	0.013	3.49	<0.001	0.04
Gender→Cat	−0.095	0.047	−2.00	0.510	-
Proximity→Cat	0.133	0.072	1.83	0.070	-
Income→Cat	0.001	0.005	0.242	0.808	-
Pat * Group Age→Cat	−0.230	0.094	−2.45	0.015	0.15
Young (N = 203)					
Pat→Cat	−0.137	0.131	−0.99	0.324	-
Elderly (N = 197)					
Pat→Cat	0.323	0.134	3.00	<0.001	0.20

Note: Pat = Passive Active Transportation, Cat = Categorical Spatial Relations, Group Age = Elderly vs. Young, MoCA = Montreal Cognitive Assessment, Education = Level of Education, Gender = Male and Female, Proximity = Residential Location, Income = Socio-Economic Status, * means interaction between reported variables.

**Table 4 ijerph-18-00709-t004:** Standardized coefficients of the moderated Model 3.

	MODEL 3				
	β	Std.Err	t	*p*-Value	Cohen’s d
Total sample (N = 400)					
Lex→Coo	1.015	0.081	12.51	<0.001	0.50
Group Age→Coo	0.389	0.073	3.87	<0.001	0.21
MoCA→Coo	0.023	0.014	1.64	0.100	-
Education→Coo	0.012	0.011	1.07	0.283	-
Gender→Coo	0.075	0.041	1.84	0.070	-
Proximity→Coo	0.069	0.059	1.17	0.240	-
Income→Coo	−0.001	0.006	−0.20	0.830	-
Lex * Group Age→Coo	−0.354	0.084	−4.22	<0.001	0.20
Young (N = 203)					
Lex→Coo	0.660	0.115	5.73	<0.001	0.35
Elderly (N = 197)					
Lex→Coo	1.400	0.118	11.70	<0.001	0.71

Note: Lex = Level of Experience, Coo = Coordinate Spatial Relations, Group Age = Elderly vs. Young, MoCA = Montreal Cognitive Assessment, Education = Level of Education, Gender = Male and Female, Proximity = Residential Location, Income = Socio-Economic Status, * means interaction between reported variables.

**Table 5 ijerph-18-00709-t005:** Standardized coefficients of the moderated Model 4.

	MODEL 4				
	β	Std.Err	t	*p*-Value	Cohen’s d
Total sample (N = 400)					
Pat→Coo	0.141	0.091	1.54	0.091	-
Group Age→Coo	0.231	0.078	2.19	0.029	0.13
MoCA→Coo	0.037	0.016	2.23	0.026	0.03
Education→Coo	0.054	0.013	3.93	<0.001	0.05
Gender→Coo	0.059	0.046	1.29	0.264	-
Proximity→Coo	0.111	0.070	1.58	0.110	-
Income→Coo	−0.001	0.005	−0.16	0.870	-
Pat * Group Age→Coo	−0.180	0.091	−2.08	0.038	0.14
Young (N = 203)					
Pat→Coo	−0.039	0.127	−0.31	0.759	-
Elderly (N = 197)					
Pat→Coo	0.321	0.130	2.46	0.014	0.20

Note: Pat = Passive Active Transportation, Coo = Coordinate Spatial Relations, Group Age = Elderly vs. Young, MoCA = Montreal Cognitive Assessment, Education = Level of Education, Gender = Male and Female, Proximity = Residential Location, Income = Socio-Economic Status, * means interaction between reported variables.

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
