# Peer review of "The Road More Travelled: The Differential Effects of Spatial Experience in Young and Elderly Participants"

_ijerph, 2021, doi:10.3390/ijerph18020709_

Round 1

Reviewer 1 Report

This research work is very interesting, with a topic that lacks relevant research. The theoretical framework is very complete and well constructed. Regarding the research method, G*Power is used, which seems to us to be very correct, to reveal the power and size of the effect, is something very positive for an investigation. The size and selection of the sample is adequate.

The authors say that a chi-square and an analysis of variance will be done, but they do not specify if the data distribution is normal or not. We do not know how reliable the data collection instruments are, nor the validity of their content or construct, although they are standardized it is good to make this explicit.

The multivariate analysis with the lambda is correct and the authors show a very timely statistical domain, which gives validity to the conclusions, my congratulations for the subject and the work done, well founded and executed.

Author Response

This research work is very interesting, with a topic that lacks relevant research. The theoretical framework is very complete and well constructed. Regarding the research method, G*Power is used, which seems to us to be very correct, to reveal the power and size of the effect, is something very positive for an investigation. The size and selection of the sample is adequate.The authors say that a chi-square and an analysis of variance will be done, but they do not specify if the data distribution is normal or not. We do not know how reliable the data collection instruments are, nor the validity of their content or construct, although they are standardized it is good to make this explicit.The multivariate analysis with the lambda is correct and the authors show a very timely statistical domain, which gives validity to the conclusions, my congratulations for the subject and the work done, well founded and executed.

Reply: Thank you for your comments. Data distribution is not normal, however the Chi-square test of independence is a non-parametric statistic, also called a distribution free test [McHugh, M. L. (2013). The chi-square test of independence. Biochemia medica: Biochemia medica, 23(2), 143-149], while ANOVA is robust under application of non- normally distributed data [Schmider, E., Ziegler, M., Danay, E., Beyer, L., & Bühner, M. (2010). Is it really robust?. Methodology]. The Italian version of the materials (i.e., MoCA, ADL, IADL, GDS, SMCQ) was used, whose references were reported. Moreover, we added the internal consistency of LPM task for categorical and coordinate components of spatial relations (please see line 388).

Reviewer 2 Report

This work presents a study on spatial representation and orientation skills of young and older people influenced by familiarity with their environment. The study is well designed and implemented. Analysis and conclusion are presented in details.

Some further questions can be considered for improving the manuscript.

  1. The keywords list doesn’t include age which is the key subject of this work.
  2. The last three sub-section numbers under section 1 are wrong.
  3. More details about the research region are expected, e.g. area, population.
  4. How were the ten well-known landmarks being selected?
  5. What do the boundaries mean in figure 2c and 2d? The labels in these two figures are not legible in the review version.
  6. How were the points determined, 28 points for each axis in line 350, and 378 points for each axis in line 355?
  7. “The Participants were classified as passive or active navigators if most of their explorations were associated with an active or passive mode of transportation”. How the “most” is defined?
  8. What does the abbreviation “LPM” mean?
  9. The popularity of mobile navigation among young people may contribute to the results to some extent, which should be addressed.
  10. The collection of information and measures of LEX and PAT shall be presented with more details.

Author Response

This work presents a study on spatial representation and orientation skills of young and older people influenced by familiarity with their environment. The study is well designed and implemented. Analysis and conclusion are presented in details.

Some further questions can be considered for improving the manuscript.

  • The keywords list doesn’t include age which is the key subject of this work. Reply: Thank you for your comment, it was done.
  • The last three sub-section numbers under section 1 are wrong. Reply: Thank you for your comment, it was corrected.
  • More details about the research region are expected, e.g. area, population. Reply: Thank you for your comment, we added some information, please see line 285-287.
  • How were the ten well-known landmarks being selected? Reply: Thank you vey much for your comment. When the LPM task was built (please see Lopez et al., 2018) the selection of the landmarks was based on a pilot aimed to rating participants’ level of familiarity and the salience of landmarks.
  • What do the boundaries mean in figure 2c and 2d? The labels in these two figures are not legible in the review version. Reply: Thank you for your comment. As emerged from the label of Figure 2: 2c) is a semi-blind map with two fixed reference points and North; 2d) is a scoring sheet: the map with the expected positions and distances. We hope the revised version will be legible.
  • How were the points determined, 28 points for each axis in line 350, and 378 points for each axis in line 355? Reply: Thank you for your comment. As emerged from the scoring method section LPM task is composed of 8 landmarks. From the categorical point of view, we compared each pair of landmarks separately on the x and y axes. If all the positions were correct, the maximum score for each axis was 28 (1 point for each correct position). From the coordinate point of view distances of each pair of landmarks were compared, separately, on the x axis and y axis. If all the distances’ comparisons were correct, the maximum score for each axis was 378 (1 point for each correct comparison).
  • “The Participants were classified as passive or active navigators if most of their explorations were associated with an active or passive mode of transportation”. How the “most” is defined? Reply: Thank you for your comment. The term “most” indicates most of the time per week. If a person used public transit or was a passenger in a car most of the time, he/she was considered like passive. We added the term weekly in the ms.
  • What does the abbreviation “LPM” mean? Reply: Thank you for your comment. We added the acronym LPM after the longer name Landmark positioning on a map (please see line 356)
  • The popularity of mobile navigation among young people may contribute to the results to some extent, which should be addressed. Reply: Thank you for your comment. It cannot be excluded that the popularity of mobile navigation among young people might contribute to the better performance of the young on coordinate, compared to the elderly (please see line 399). However, even if this was true, it would not explain the comparable performance of the young and the elderly with a high level of LEX and PAT on coordinate.
  • The collection of information and measures of LEX and PAT shall be presented with more details. Reply: Thank you for your comment, we added information about data collection as well as measures of LEX and PAT (please see line 299-337).

Reviewer 3 Report

The study is well designed. The obtained effects of the frequency and mode of exploration on the accuracy of determining the location of a landmark are hardly surprising. Of interest, however, are the observed effects of the interaction of these independent variables with the age of the subjects. The following questions arise regarding the text of the article:

  • Line 106: “thee shopping”. Is this not a typo?

  • Lines 495-496: “With respect to the effect of age on categorical and coordinate spatial relations, young and 496 elderly participants showed a significant but small difference in categorical performance”. However, in Table 1 this difference is marked as non-significant.

And one recommendation: it would be nice if the interactions (Lex * Group Age → Cat, etc.) were depicted in figures.

Author Response

The study is well designed. The obtained effects of the frequency and mode of exploration on the accuracy of determining the location of a landmark are hardly surprising. Of interest, however, are the observed effects of the interaction of these independent variables with the age of the subjects. The following questions arise regarding the text of the article:

  • Line 106: “thee shopping”. Is this not a typo? Reply: Thank you, it was correct.
  • Lines 495-496: “With respect to the effect of age on categorical and coordinate spatial relations, young and 496 elderly participants showed a significant but small difference in categorical performance”. However, in Table 1 this difference is marked as non-significant. Reply: Thank you for your comment. This allowed us to clarify the results related to age differences. In the reviewed version of the ms., we have specified that the sentence included in line 528-530 was referred to the moderated models.
  • And one recommendation: it would be nice if the interactions (Lex * Group Age → Cat, etc.) were depicted in figures. Reply: Thank you for your comment, we added four figures related to the interactions in supplementary materials.

Reviewer 4 Report

I found the reviewed text of an excellent quality that significantly contributes to the field of spatial perceptions and sketch maps, especially the influence of age and familiarity with the area. Therefore, I recommend the publication of the paper in the Special Issue of the International Journal of Environmental Research and Public Health. The paper is focussed on the research issue, is well structured and has a solid methodological and theoretical body. Moreover, I appreciate that the authors worked with a wide scale of related literature.

However, I have some recommendations for the revision:

1) Abstract – to make readers more familiar with the issue of the paper (and to promote the paper) I would recommend restructuring the abstract – especially I would suggest leaving the "headings" out of the abstract (i.e. delete Background:, Methods:, Results:, Conclusions:) and rewriting it as a continuous flowing text.

Moreover, the text introduced as "Methods:" explains the study's aims and general focus rather than its methodology.

2) Introduction – I will welcome if authors could explicitly express the overall intentions of authors/focus and aims of the study directly at the beginning of the Introduction part. That could help attract readers and make the paper more understandable from its beginning – e.g. introduce the problem, explain why it should be researched, and how it contributes to the expansion of knowledge in the field.

It is currently mentioned too late – on page 5 (these are correctly explained detailed aims and hypotheses, but the general ail should be introduced earlier).

3) A systematic review into factors influencing sketch map quality could help authors set the study into the broader context of the spatial-perception studies and discuss the influence of familiarity and the age on the spatial perceptions. It could be beneficial to work with its findings in the 1.2 and 1.4 sections.

Hátlová, K.; Hanus, M. A Systematic Review into Factors Influencing Sketch Map Quality. ISPRS Int. J. Geo-Inf. 2020, 9, 271. https://doi.org/10.3390/ijgi9040271

4) 1. Participants – please, state explicitly the grade of university students participating in the study – as the length of their studies could influence the general familiarity with the area (students in their 1st year vs students in more advanced years).

5) 2. Materials and Procedure – the majority of the text relates to participants and the inclusion criteria. It should be, therefore, mentioned instead in subsection 2.1

6) I appreciate, that authors are aware of their findings' limitations and discuss them explicitly in the Discussion section. However, I am missing a discussion of the influence of various levels of map skills of participants on the results of the study – specifically, as participants were asked to create a sketch map (requiring a certain level of map-drawing skills), these sketch maps undoubtedly varied not only according to spatial perception of the area (and the familiarity, way of transport etc.) but surely according to the level of map skills of participants. This influence should be discussed among the limits of the study.

7) Conclusions – some parts of the text repeat the findings mentioned in the Discussion section (especially the second paragraph of the Conclusion section) and are therefore redundant.

Minor typos:

p. 1, line 39: … experience [7]. – missing dot

p. 3, line 106: doing the shopping -

p. 3, line 140: downtown Sydney, Australia – i/y

p. 4–5, 1. Introduction: sub-sections numbering – change 2.3, 2.4 and 2.5 to 1.3, 1.4, and 1.5

Author Response

I found the reviewed text of an excellent quality that significantly contributes to the field of spatial perceptions and sketch maps, especially the influence of age and familiarity with the area. Therefore, I recommend the publication of the paper in the Special Issue of the International Journal of Environmental Research and Public Health. The paper is focussed on the research issue, is well structured and has a solid methodological and theoretical body. Moreover, I appreciate that the authors worked with a wide scale of related literature.

However, I have some recommendations for the revision:

  • Abstract – to make readers more familiar with the issue of the paper (and to promote the paper) I would recommend restructuring the abstract – especially I would suggest leaving the "headings" out of the abstract (i.e. delete Background:, Methods:, Results:, Conclusions:) and rewriting it as a continuous flowing text. Moreover, the text introduced as "Methods:" explains the study's aims and general focus rather than its methodology. Reply: Thank you for your comment. We deleted the headings and we added a sentence to explain the method (please see line 14-26).
  • Introduction – I will welcome if authors could explicitly express the overall intentions of authors/focus and aims of the study directly at the beginning of the Introduction part. That could help attract readers and make the paper more understandable from its beginning – e.g. introduce the problem, explain why it should be researched, and how it contributes to the expansion of knowledge in the field. It is currently mentioned too late – on page 5 (these are correctly explained detailed aims and hypotheses, but the general ail should be introduced earlier). Reply: Thank you for your comment. At the beginning of the introduction, we added our intentions and focus (please see line 44-63).
  • A systematic review into factors influencing sketch map quality could help authors set the study into the broader context of the spatial-perception studies and discuss the influence of familiarity and the age on the spatial perceptions. It could be beneficial to work with its findings in the 1.2 and 1.4 sections. Hátlová, K.; Hanus, M. A Systematic Review into Factors Influencing Sketch Map Quality. ISPRS Int. J. Geo-Inf. 2020, 9, 271. https://doi.org/10.3390/ijgi9040271 Reply: Thank you for your comment, it was an interesting paper, very useful for our purpose. We cited it in the ms. and in the reference list.

  • Participants – please, state explicitly the grade of university students participating in the study – as the length of their studies could influence the general familiarity with the area (students in their 1st year vs students in more advanced years). Reply: Thank you for your comment. Our students were at the first year of master’s degree. We added this information in the ms. (please see line 321-323)
  • Materials and Procedure – the majority of the text relates to participants and the inclusion criteria. It should be, therefore, mentioned instead in subsection 2.1 Reply: Thank you for your comment, it was done.

  • I appreciate, that authors are aware of their findings' limitations and discuss them explicitly in the Discussion section. However, I am missing a discussion of the influence of various levels of map skills of participants on the results of the study – specifically, as participants were asked to create a sketch map (requiring a certain level of map-drawing skills), these sketch maps undoubtedly varied not only according to spatial perception of the area (and the familiarity, way of transport etc.) but surely according to the level of map skills of participants. This influence should be discussed among the limits of the study. Reply: Thank you for your comment. As emerged from the description of the task, participants were required only to pinpoint the eight landmarks. This kind of sketch map is a very simple version, that unlike the classic sketch maps, it does not contain all the elements such as paths, edges, districts, nodes, and landmarks, but just landmarks. So, in order to complete the semi-blind map the level of map-drawing skills is not an important variable to consider. For this reason we think that it is not necessary to mention this aspects as a limitation. However, we remain open to your further suggestions.

  • Conclusions – some parts of the text repeat the findings mentioned in the Discussion section (especially the second paragraph of the Conclusion section) and are therefore redundant.Reply: Thank you for your comment. We decided to delete some parts of the second paragraph of the conclusion.

Minor typos:

  1. 1, line 39: … experience [7]. – missing dot Reply: Thank you, it was done.
  2. 3, line 106: doing the shopping - Reply: Thank you, it was done.
  3. 3, line 140: downtown Sydney, Australia – i/y Reply: Thank you, it was done.
  4. 4–5, 1. Introduction: sub-sections numbering – change 2.3, 2.4 and 2.5 to 1.3, 1.4, and 1.5 Reply: Thank you, it was done.

Reviewer 5 Report

The paper addresses a very interesting topic, but I think will benefit from a deeper description of the measures used. Please see the following specific points:

  • Abstract. I suggest revising the abstract focusing on the core of the study. For instance, I think the background sentence is too generic. Then, I do not understand how the results presented in the abstract (and then in the paper) lead you to conclude that elderly benefit more from exploring their hometown, I suggest to rephrase or explain.
  • Spatial relations. I suggest adding a description (and maybe an example) of what do you mean by “spatial relations” when introducing this term for the first time in the introduction. This notion is then central in the paper and needs a deeper explanation based on the literature. For instance, in the Results and Discussion you mention Positions and distances, I think this should be clear related to categorical and coordinate categorization from the beginning. 
  • Related to the above point, you extrapolated the central measures (categorical and coordinate) from a sketch map task. This should be at least briefly introduced in the background literature, i.e. what is already known about this task in young and older adults? And in particular, in its relationship with familiarity with the environment.
  • You used frequency and mode of exploration as predictors. What is the rationale of this? Something is present in the Introduction, but I think a more specific reference of this should be added (or rephrased).
  • Participants, materials and procedure. I found a bit confusing reading those paragraphs. I suggest adding all the inclusion criteria in the Participants section. And  to present all the materials and then the procedure, the latter including a clear presentation of the task and their order of presentation in the session
  • Differences in the education level, socioeconomic status, and age groups, could these variables be confounded? 
    1. Maybe I did not understand completely, but you inferred the socio-economic status based on the place where people live? If it is true, I see a confound with residential locations. 
    2. Related to this, you stated in the text that young people had more income than older adults, is this confounding with the age group? However, in the Table you reported a non significant relationship, please clarify this point.
    3. I suggest adding a correlation table between all variables
  • Do you find the same significant interactions between passive  transportation and level of experience and age group on categorical and coordinate measures, at the net of the other predictors using nested/hierarchical models?
  • Why do young women report such a low level of experience with the environment? Are they not familiar with the environment itself, maybe are they living there for a fewer number of years?
  • What about the familiarity with each of the landmarks presented in the sketch map task? If I frequent it a lot maybe I am more able to position it on a map and more accurate with the distances with the other landmarks. I suggest adding not having this measure as a limitation altogether with the limitation about familiarity already present in the discussion.

Minor points

  • Paragraph 1.2. I suggest to rephrase the initial sentence ”Our knowledge of space changes across the life span” to make it clearer or at least to add a reference. 
  • Row 92. Parenthesis missing
  • In tables: “<.05” please substitute with the actual value of p
  • Row 468. Wrong parenthesis
  • B are reported in table, maybe standardized betas should help a reader
  • Discussion: automatic softwares for map drawing tasks already exists (e.g., Gardony, A.L., Taylor, H.A., & Brunyé, T.T. (2016). Gardony map drawing analyzer: Software for quantitative analysis of sketch maps. Behavior Research Methods, 48(1), 151 - 177)

Author Response

The paper addresses a very interesting topic, but I think will benefit from a deeper description of the measures used. Please see the following specific points:

  • I suggest revising the abstract focusing on the core of the study. For instance, I think the background sentence is too generic. Then, I do not understand how the results presented in the abstract (and then in the paper) lead you to conclude that elderly benefit more from exploring their hometown, I suggest to rephrase or explain. Reply: Thank you for your comment. We have rephrased the abstract. We take this opportunity to better explain what we mean with the last two sentences in the abstract (Ceteris paribus, the role of the direct experience with exploring their hometown on spatial mental representations appeared to be more important in the elderly than in the young. This advantage appears to make the elderly wiser and likely protects them from the detrimental effects of aging on spatial mental representations). If on the one hand the elderly with a low level of LEX and PAT reported a worse performance than the young with the same low level of direct experience with the environment, on the other hand the elderly with a high level of LEX and PAT do not show this difference with the young. In other words, the direct experience with the environment has a stronger effect on spatial mental representations among the elderly than the young. For these reasons we conclude that “the direct experience with exploring their hometown on spatial mental representations appeared to be more important in the elderly than in the young” as well as the “advantage appears to make the elderly wiser and likely protects them from the detrimental effects of aging on spatial mental representations”.
  • Spatial relations. I suggest adding a description (and maybe an example) of what do you mean by “spatial relations” when introducing this term for the first time in the introduction. This notion is then central in the paper and needs a deeper explanation based on the literature. For instance, in the Results and Discussion you mention Positions and distances, I think this should be clear related to categorical and coordinate categorization from the beginning. Reply: Thank you for your comment, it was done (please see line 81 and line 82).
  • Related to the above point, you extrapolated the central measures (categorical and coordinate) from a sketch map task. This should be at least briefly introduced in the background literature, i.e. what is already known about this task in young and older adults? And in particular, in its relationship with familiarity with the environment. Reply: Thank you for your comment. We added in the introduction a sentence related to a paper newly published that resumes the results of the most recent research (please see line 183-187).
  • You used frequency and mode of exploration as predictors. What is the rationale of this? Something is present in the Introduction, but I think a more specific reference of this should be added (or rephrased). Reply: Thank you for your comment. We have dedicated the paragraph 1.3 to explain why we go to use frequency and mode of exploration as new variables to measure familiarity with the environment. Moreover, our conceptualization is supported by a recently published paper added in the manuscript (Hátlová & Hanus, 2020) in which the role of mobility, familiarity and the other factors that influence the relation between familiarity and spatial mental representations were taken into account in the sketch map performance.

  • Participants, materials and procedure. I found a bit confusing reading those paragraphs. I suggest adding all the inclusion criteria in the Participants section. And to present all the materials and then the procedure, the latter including a clear presentation of the task and their order of presentation in the session. Reply: Thank you for your comment, it was done.

  • Differences in the education level, socioeconomic status, and age groups, could these variables be confounded? Reply: Thank you for your comment. In this study a clear definition of the abovementioned variables was reported. The socioeconomic status (line 330-331) does not take into account the educational level. Anyway, it is plausible that these variables could be associated to each other, but this was controlled in the analyses. For this reason, their effects are not confounded (please also see the correlation matrix in supplementary materials).

  • Maybe I did not understand completely, but you inferred the socio-economic status based on the place where people live? If it is true, I see a confound with residential locations. Related to this, you stated in the text that young people had more income than older adults, is this confounding with the age group? However, in the Table you reported a non significant relationship, please clarify this point. Reply: Thank you for your comment. As explained in line 326-330, the “residential location” is based on proximity to the downtown area. While the socio-economic status derived from rental costs or property values in their neighbourhood. Also in this case, these two measures could be associated to each other, but this was controlled in the analyses.The sentence in the ms. regarding the income was not correct. We apologise for this oversight, we corrected it.
  • I suggest adding a correlation table between all variables. Reply: Thank you very much for your comment. We added the correlation matrix in supplementary materials

  • Do you find the same significant interactions between passive transportation and level of experience and age group on categorical and coordinate measures, at the net of the other predictors using nested/hierarchical models? Reply: Thank you for your comment. We have not used nested/hierarchical models, but four independent standard moderated models net of residential zones, socio-economic status, gender, education, and cognitive efficiency/intelligence used as covariates. The use of a standard model is more conservative than a nested/hierarchical model.
  • Why do young women report such a low level of experience with the environment? Are they not familiar with the environment itself, maybe are they living there for a fewer number of years? Reply: Thank you for your comment. As emerged from table 1 the Cochran-Mantel-Haenszel test was not significant between female and male participants.

  • What about the familiarity with each of the landmarks presented in the sketch map task? If I frequent it a lot maybe I am more able to position it on a map and more accurate with the distances with the other landmarks. I suggest adding not having this measure as a limitation altogether with the limitation about familiarity already present in the discussion. Reply: Thank you for your comment. We added a sentence in the limitations.

Minor points

Paragraph 1.2. I suggest to rephrase the initial sentence ”Our knowledge of space changes across the life span” to make it clearer or at least to add a reference.

  1. Reply: Thank you for your comment, it was deleted.

Row 92. Parenthesis missing

  1. Reply: Thank you, it was done.

In tables: “<.05” please substitute with the actual value of p

  1. Reply: Thank you for your comment, it was done.

Row 468. Wrong parenthesis

  1. Reply: Thank you, it was done.

B are reported in table, maybe standardized betas should help a reader

  1. Reply: Thank you very much for your comment, it was done.

Discussion: automatic softwares for map drawing tasks already exists (e.g., Gardony, A.L., Taylor, H.A., & Brunyé, T.T. (2016). Gardony map drawing analyzer: Software for quantitative analysis of sketch maps. Behavior Research Methods, 48(1), 151 - 177)

  1. Reply: Thank you for your comment. We know that Gardony developed a software for sketch maps, but it does something different than what we want to do. As emerged from the ms., we want to develop a software that can automatically code measures for the distinct analysis of positions and distances between points in a geographic space.

Round 2

Reviewer 5 Report

The authors have done a good job in addressing most of my earlier concerns and suggestions. I do not have any further comments.